# Supervisor-Worker Problems with an Application in Education

**DOI:** 10.3390/s21061965

**Published:** 2021-03-11

**Authors:** Dorin Shmaryahu, Kobi Gal, Guy Shani

**Affiliations:** Department of Software and Information Systems Engineering, Ben Gurion University of the Negev, Beer-Sheva 84105, Israel; shanigu@bgu.ac.il

**Keywords:** e-learning, education, interactive learning, artificial intelligence, choice, planning, multi agent planning, classical planning

## Abstract

In many e-learning settings, allowing students to choose which skills to practice encourages their motivation and contributes to learning. However, when given choice, students may prefer to practice skills that they already master, rather than practice skills they need to master. On the other hand, requiring students only to practice their required skills may reduce their motivation and lead to dropout. In this paper, we model this tradeoff as a multi-agent planning task, which we call SWOPP (Supervisor- Worker Problem with Partially Overlapping goals), involving two agents—a supervisor (teacher) and a worker (student)—each with different, yet non-conflicting, goals. The supervisor and worker share joint goals (mastering skills). The worker plans to achieve his/her own goals (completing an e-learning session) at a minimal cost (effort required to solve problems). The supervisor guides the worker towards achieving the joint goals by controlling the problems in the choice set for the worker. We provide a formal model for the SWOPP task and two sound and complete algorithms for the supervisor to guide the worker’s plan to achieve their joint goals. We deploy SWOPP for the first time in a real-world study to personalize math questions for K5 students using an e-learning software in schools. We show that SWOPP was able to guide students’ interactions with the software to practice necessary skills without deterring their motivation.

## 1. Introduction

When using e-learning software for practicing skills, students can often choose which of several tasks to complete [1]. Research in the social sciences has demonstrated that providing people with different task choices increases their motivation and task performance [2]. For example, allowing students to choose which problem to solve from a candidate set was shown to positively contribute to their learning [3]. On the other hand, students may choose to solve problems that are easy and familiar, not venturing outside of their “comfort zone” to try more challenging problems that increase learning [1]. There is, thus, a need for e-learning systems to personalize the choice set of problems in a way that achieves teachers’ learning goals while maintaining students’ motivation.

We model this tradeoff as a multi-agent planning problem [4] called SWOPP (Supervisor- Worker Problem with Partially Overlapping goals) with two agents: a supervisor (teacher) and a worker (student). The worker can apply a sequence of actions in order to achieve its goals, and executing actions incurs a cost. The worker can control which actions to choose from a given choice set, while the supervisor controls the action costs. The supervisor must guide the worker towards achieving the supervisor’s goals, while still achieving the goals of the worker, and without reducing its set of choices. The supervisor and the worker are not fully collaborative, having different goals, yet not strictly competitive, as their goals do not conflict.

The supervisor can modify the worker’s cost function, by choosing to increase the cost of worker actions. Thus, the supervisor can make certain worker actions less desirable for a worker, without removing them from the set of actions under consideration. The worker can always choose from the original set of actions, but undesirable choices to the supervisor are made less attractive for the worker. The worker plans its actions using the revised cost function that is dictated by the supervisor.

In the e-learning setting, the student goal is to successfully complete a session with the system. The goal of the teacher is for the student to practice certain skills. There are several possible solution paths to completing the e-learning session. Each solution path contains a set of problems and their associated skills. The student is free to choose any of these solution paths. The cost function of the student measures the effort that is required to solve problems (lower costs are associated with easier problems for the student). The teacher can control the solution paths, allowing to personalize possible solution paths for each student. For example, the teacher can guide the student towards a solution path with new skills by increasing the number of problems in a solution path which contains skills that the student already knows, making this easier solution path less attractive to the student.

The cost for the teacher represents the amount of work that the teacher must invest to create a longer path, that requires more effort from the student. For example, the teacher can create additional questions to be added to the path. The teacher wishes for the student to acquire a new skill, without overloading or frustrating the student. Hence, the teacher will guide the student to the path that minimizes the student cost in achieving both teacher and student not conflict goals.

While, in this paper, we focus on an education application, SWOPP is a general model that can be applied to other problems. For example, a city that wishes to persuade drivers to avoid school areas when the students arrive at school or leave. SWOPP can compute the necessary speed limit for each driver to make driving near the school undesirable. Another application for SWOPP is to regulate traffic during rush hours in toll roads. SWOPP can design a personalized toll for each driver, sufficiently high to persuade the driver to take other roads.

Our research problem is to define and solve the SWOPP problem. We provide two algorithms that receive as input the goals and cost functions for both the worker and the supervisor. Both algorithms output a revised cost function for the worker that produces an optimal plan for the worker (minimizes its costs) while fulfilling the goals for both supervisor and worker. The first algorithm (called Cost Function Modification (CFM)) is a brute force approach guaranteed to minimize the modification costs for the supervisor, but requires to generate an exponential number of plans for the worker. The second algorithm (called Incremental CFM (ICFM)) is an iterative approach that may in practice traverse far fewer plans, but does not guarantee optimality for the supervisor.

We apply SWOPP in the wild by using it to personalize math questions from different topics to K5 students using e-learning software in elementary schools. We developed a game, where the player advances on a path to a goal by solving math problems. The player can choose from different paths, where each path requires the player to answer questions associated with different math topics. Once the player solved a pre-defined number of questions, thus demonstrating her mastery of a skill, a gate was opened allowing her to advance the path towards the goal. SWOPP was used to set the number of gates that must be opened at each path, according to a specific skill, set by the teacher for that particular student. This is the first time SWOPP was formalized and has been used in educational settings.

We formulate an SWOPP problem for this setting, defining a personal cost function for each student. We show that more students practiced their necessary skills when using our approach, even though they were given free choice about which skills to practice. These students solved more problems correctly than other students whose sessions were not tailored by SWOPP and reported a high perceived level of choice.

This paper provides three contributions: (1) the formalization of the supervisor-worker SWOPP problem; (2) two sound and complete algorithms for solving the problem; and (3) applying SWOPP in a real world education setting and validating our approach in this setting.

## 2. Related Work

Several approaches within AI and education have used computational methods for personalizing learning paths of students. Recommendation algorithms have been used to personalize educational content to students by aggregating the rankings of similar students [5,6]. Other researchers used graphical models or neural nets to sequence content for students by inferring their skill acquisition (i.e., mastery level) over time given their performance levels on different question sequences [7,8]. The models are trained on large data sets from multiple students using machine learning algorithms that account for latent variables. Other methods have considered the exploration/exploitation tradeoff when generating contents to students by using Multi Armed Bandits [5,9], or through knowledge transfer models [10].

Our approach is inspired by recent studies in behavior change and persuasion that show people prefer to be able to choose between several alternatives [11,12]. It is widely agreed in psychology and economics research that the provision of choice increases the intrinsic motivation, perceived control, task performance, and overall satisfaction of people [13]. Although choosing between too many options can be confusing, choosing between a small set of options is known to be preferred to no choice [2]. This was also shown in the educational domain [1,3].

SWOPP relates to the environment design problem in AI in which an interested party, the system designer, can make changes to the environment of an agent in order to encourage desired behavior [14].  Keren et al. [15] model environment design settings in which the system designer and the worker optimize the same utility function. The designer attempts to introduce changes to the environment that would reduce the cost of achieving the joint goals. This is very different than the SWOPP model, where the supervisor increases the worker cost for some of its actions, in order to achieve the supervisor goals. Keren et al. allow modifications to the transition probabilities, while we only increase the cost of worker actions, which is more suitable in some real world scenarios.

Keren et al. [16] studied the goal recognition design (GRD) problem, where the environment can be modified to allow the system designer to elicit the goals of the agent. GRD takes a similar modeling approach to ours, based on classical planning. However, in GRD, the system designer cannot increase the overall cost of the agent’s plan, while we must increase this cost in order to meet the goals of the designer. In addition, GRD allows only action removal by the system designer, which is equivalent to increasing the cost of actions to infinity, while we allow more subtle cost increases.

Our work is also related to the inverse optimization problem in the operations research community [17], inter alia. The input to an inverse optimization problem is a solution and a cost function. The output of the problem is a revised cost function such that the given solution would be optimal. Inverse optimization was studied for several applications, such as constraint optimization and auctions, and has been applied to domains, such as assembly lines and health care. SWOPP combines insights from both inverse optimization and environment design.

Reward shaping is a form of reinforcement learning that modifies the original rewards in MDPs [18], to provide domain information to the learning process, to control its exploration, and to inject human input [19]. Reward shaping differs from SWOPP in that it encourages faster learning, rather than directing the agent to different goals.

De Giacomo et al. [20] describe an online process where a supervisor limits the set of actions available to a worker at each step of the execution. The goals of the worker and the supervisor may conflict, and the supervisor must prevent the worker from behavior that is deemed undesirable by the supervisor. They differ from our work in guiding the worker away from undesirable behavior, rather than guiding it towards achieving additional goals. They focus, as we do, on reducing the restrictions on the choices of the worker. They take a logic-based approach, ignoring action costs.

Our work is also related to the incentive design task [21,22], where a supervisor can design incentives for an agent to achieve a desired action. Specifically, Chen et al. [23] consider a setting where the agent uses a multi-armed bandit model to choose actions, but the supervisor does not know the beliefs of the agent about the action values. Assuming that the given costs represent the minimal required effort, we do not allow to reduce the costs of the worker actions with respect to its original cost function.

## 3. Background: Classical Planning

We provide a necessary background for the SWOPP model. A *classical planning* problem is a tuple $\langle P,A,s_{I},G,C\rangle$, where *P* is a set of propositions, *A* is a set of actions, $s_{I}\subset P$ is the initial state—a set of propositions that are initially true, and G⊂P is the set of goal propositions that the agent must achieve. The propositions in *P* describe the possible world states, such that each world state *s* is defined by a truth or false assignment to all propositions in *P*.

An action a∈A is a pair, {*pre*(*a*), *effects*(*a*)}, where *pre*(*a*) is a conjunction of literals, denoting a set of preconditions that must hold before *a* can be applied. We say that *a* is applicable in a state *s* if s⊧pre(a). The action effect *effects*(*a*) is a conjunction of literals, modeling the change to the world following the execution of *a*. We use a(s) to denote the state that is obtained after *a* is executed in state *s*.

A solution to a classical planning problem is a sequence of actions $\pi =\langle a_{1},\ldots ,a_{n}\rangle$, such that $\pi \left(i\right)=a_{i}$ is applicable in the state $a_{i-1}\left(\ldots \left(a_{1}\left(s_{I}\right)\right)\right)$, and $G\subset a_{n}\left(a_{n-1}\left(\ldots \left(a_{1}\left(s_{I}\right)\right)\right)\right)$. Applying the plan sequentially, starting from the initial state $s_{I}$, will result in a valid state where all goal propositions hold. There are many approaches to constructing classical planners for computing solution plans (see, e.g., Hoffmann [24], Helmert [25]).

A *time-based* (non-stationary) cost function for an agent C:A×T→R assigns a numerical cost to the execution of an action by the worker at a given time. The worker cost for executing plan π is $C\left(\pi \right)=\sum_{t=0,\ldots ,|\pi |}c\left(\pi \left(t\right),t\right)$, where π(t) is the action dictated by the plan at time *t*, and c(a,t) is the cost associated with action *a* at time *t*.

## 4. The SWOPP Problem

We now define the SWOPP problem under full observability and deterministic actions. We begin with motivating applications, highlighting the main properties of the problem, and focusing on challenges that must be handled.

**Definition** **1.***A* SWOPP *problem is a tuple*
$\langle P,A,s_{I},G^{W},G^{S},C_{0},C_{s}\rangle$*, where P is a set of propositions; A is a set of actions for the worker;*
$s_{I}$
*is an initial state;*
$G^{W}$
*and*
$G^{S}$
*(subsets of P) are the set of goals of the worker and supervisor, respectively;*
$G^{S}$ and $G^{W}$
*are consistent in the sense that there exists a reachable state s such that*
$s⊧G^{S}\cup G^{W}$; $C_{0}$
*is the initial cost function for the worker, representing the minimal possible effort for the worker for applying actions.*
$C_{s}$
*is the supervisor cost function*
$C_{s}\left(a,t,c,c^{\prime }\right)$
*assigning a cost for increasing the worker’s cost function for action a at time t from c to*
$c^{\prime }$.

The supervisor’s objective is that the worker achieves both $G^{W}$ and $G^{S}$, while the worker’s objective is to achieve only $G^{W}$. At each step, the worker takes the optimal action for achieving its goals, given its time-based cost function. The supervisor can guide the worker towards performing actions that are desirable to the supervisor, by modifying the costs of other actions for the worker at specific times. We allow only modifications that increase the worker cost beyond the initial cost $C_{0}\left(a,t\right)$, which we assume to be the minimal effort required to execute *a* at *t*. An increase to the worker’s cost function incurs a cost to the supervisor. We assume a worker that minimizes the sum of action costs. The supervisor must minimize the worker’s costs.

**Definition** **2.***A plan*$\pi =\langle a_{1},\ldots ,a_{n}\rangle$*is* SW-optimal *for a worker cost function C if it satisfies the goals*
$G^{S}$
*and*
$G^{W}$
*and is optimal for the worker cost function C.*

A solution for the supervisor-worker problem (SWOPP) is a worker cost function $C^{\prime }$ such that any plan $\pi^{\prime }$ that satisfies $G^{W}$ and is optimal with respect to $C^{\prime }$ is also SW-optimal for $C^{\prime }$. The optimal worker plan under the cost function outputted by an algorithm must complete the goals of both worker and supervisor.

The supervisor can change the worker cost for an action *a* at any time *t* from the current cost *c* to any other cost $c^{\prime }$, as long as $c^{\prime }\geq C_{0}\left(a,t\right)$, i.e., the cost can never be lower than the value specified by the initial cost function. The cost modifications are intended to guide the worker away from the actions with the revised costs. The supervisor must, however, pay a cost for modifying the worker’s cost function from its original value. This additional cost can reflect the worker’s discontent from the cost increase, or the physical cost to the supervisor in modifying the worker’s cost function. Given a revised cost function $C^{\prime }$ and an SW-optimal plan $\pi^{\prime }$ for $C^{\prime }$, the supervisor total cost
(1)$$C_{s}^{+}=\sum_{t=0,\ldots ,|\pi^{\prime }|,a\in A}C_{s}\left(a,t,C_{0}\left(a,t\right),C^{\prime }\left(a,t\right)\right),$$
that is, the sum of the cost of deviations from the initial cost function $C_{0}$.

Note that, while SW-optimal plans under $C^{\prime }$ satisfy the goals of the supervisor, they do not guarantee minimal modification costs for the supervisor. To address this, we make use of the definition below.

**Definition** **3.***An optimal solution cost function (OSCF) for SWOPP is a revised worker cost function*$C^{\prime }$*, such that (1) any plan*$\pi^{\prime }$*that is optimal for the worker (i.e., minimizes*$C(\pi^{\prime })$*) is also SW-optimal for*$C^{\prime }$; and (2) $C^{\prime }$*is optimal for the supervisor (i.e., minimizes the supervisor cost*$C_{s}^{+}$*for* all *possible SW-plans given the initial cost function*
$C_{0}$*).*

### Running Example

We use the following running example to illustrate the SWOPP problem and our approach.

**Example** **1.** 

Consider the navigation problem in Figure 1. In this domain, a worker’s task is to move from one location to another. Edges are undirected, i.e., the worker can move in both directions over an edge. The action costs for moving between two connected cells represent the movement difficulty. This example is analogous to an education setting. Nodes represent skills; edges between nodes represent math questions; arriving at a node means solving a math question and achieving the skill associated with the node. The goal of the student is to reach $n_{g}$, which represents finishing the exercise, by any possible path. The goal of the teacher is for the student to learn specific skills, represented by particular nodes. A student can pick a shorter path, allowing him to achieve a skill with less questions and, hence, less effort, but the questions may be harder than the questions on a longer path.

The worker is positioned initially in cell $n_{0}$ and has 3 available actions: (1) move through a high difficulty (hard) connection to cell $n_{1}$, incurring a cost of 6, (2) move through a low difficulty (easy) connection to cell $n_{2}$ incurring a cost of 1, or (3) move through a moderate connection to cell $n_{3}$ for a cost of 3.

The set of propositions *P* models the difficulty level of the possible connections between the cells, denoted by (*easy*
$n_{i}$$n_{j}$),(*medium*
$n_{i}$$n_{j}$), (*hard*
$n_{i}$$n_{j}$), the current location of the worker, (*loc*
$n_{i}$), and the visited cells denoted by (*visited*
$n_{i}$). An action $move(n_{i},n_{j})$ represents moving on a connection from $n_{i}$ to $n_{j}$. The precondition for all moving actions is for the worker to be in cell $n_{i}$ and an available connection to be at some difficulty level between $n_{i}$ and $n_{j}$. The location of the worker at cell $n_{j}$ is denoted by (*loc*
$n_{j}$).

The worker goal is to be at $n_{g}$, i.e., $G^{W}$ is (*loc*
$n_{g}$). The supervisor goal, $G^{S}$, is for the worker to visit $n_{2}$ and $n_{3}$, which is (*visited*
$n_{2}$) and (*visited*
$n_{3}$). The initial time-based cost function $C_{0}$ sets costs of 1, 3, and 6, for moving through easy, medium, and hard connections, respectively, for all time t=0,1,…,∞. Obviously, the union of $G^{W}$ and $G^{S}$ can be achieved, for example, by the plan: $\pi_{s}=\langle move\left(n_{0},n_{3}\right),move\left(n_{3},n_{4}\right),move\left(n_{4},n_{2}\right),move\left(n_{2},n_{4}\right),move\left(n_{4},n_{g}\right)\rangle$.

The supervisor cost function $C_{s}$ is derived directly from the cost increase for the worker, i.e., $C_{s}\left(a,t,C\left(a,t\right),C^{\prime }\left(a,t\right)\right)=C^{\prime }\left(a,t\right)-C\left(a,t\right)$. For example, if $C(move\left(n_{0},n_{2}\right),0)=1$ and the supervisor increases the cost of by 2, i.e., $C^{\prime }\left(move\left(n_{0},n_{2}\right),0\right)=3$, then $C_{s}\left(a,t,3,5\right)=2$.

A possible solution for this SWOPP problem is the following revised cost function: $C^{\prime }\left(move\left(n_{0},n_{2}\right),0\right)=10,C^{\prime }\left(move\left(n_{0},n_{1}\right),0\right)=12,C^{\prime }\left(move\left(n_{4},n_{g}\right),2\right)=5,C^{\prime }\left(move\left(n_{0},n_{2}\right),2\right)=10$. Planning under this modified cost function yields the SW-optimal plan $\pi_{s}$. However, the supervisor cost, $C_{s}^{+}=\left(10-1\right)+\left(12-6\right)+\left(5-1\right)+\left(10-1\right)=28$, is not minimal. Hence, this solution is not an OSCF.

An OSCF solution to this SWOPP problem is the following revised cost function: $C^{\prime }\left(move\left(n_{0},n_{1}\right),0\right)=6,C^{\prime }\left(move\left(n_{4},n_{g}\right),2\right)=4,C^{\prime }\left(move\left(n_{1},n_{g}\right),1\right)=4$. Planning under this modified cost function yields an SW-optimal plan. The supervisor cost, $C_{s}^{+}$ = 11, is minimal, and this solution is OSCF.

## 5. Solving the SWOPP Problem

In this section, we provide two algorithms for computing solutions to SWOPP. The first provides an OCSF solution but needs to enumerate all possible worker plans. We also present an alternative incremental approach for outputting a solution that may traverse fewer plans but does not guarantee to output an OSCF solution. Both approaches leverage a black box classical planner, WorkerPlan. This planner receives as input a classical planning problem and outputs a minimal cost plan for the problem.

### 5.1. CFM: Brute Force Approach

The first approach, called Cost Function Modification (CFM) increases the costs of all possible plans with lower worker cost than the SW-optimal plan.

We begin with some notation. Let SupervisorPlans be a black box classical planner for computing $\Pi_{s}$—the set of all possible SW-optimal plans for an initial cost function $C_{0}$. By Definition 2, each plan in $\Pi_{s}$ is optimal for the worker and has the same minimal cost, which we denote $C^{*}$. Computing $C^{*}$ can be done by running WorkerPlan over a planning problem, where the goal is to obtain both $G^{W}$ and $G^{S}$. In addition, $G^{S}$ must be obtained before $G^{W}$, as if the worker achieves $G^{W}$ before $G^{S}$, it has no incentive to continue working. Let WorkerPlans be black box classical planner for computing the set $\Pi_{w}$ of all possible plans that (1) achieve the worker goals $G^{W}$ but not the supervisor goals $G^{S}$ and (2) have a worker cost less or equal to $C^{*}$.

Given a supervisor plan $\pi_{s}\in \Pi_{s}$, we define a linear program (LP) that receives as input the worker cost $C^{*}$ for $\pi_{s}$, all worker plans in $\Pi_{w}$, the initial cost function $C_{0}$, and the supervisor cost function $C_{s}$. The goal of the LP $LP(\Pi_{w},C^{*},\pi_{s},C_{0},C_{s})$ is to increase the cost of all worker plans in $\Pi_{w}$ to be above $C^{*}$, making $\pi_{s}$ preferable to any plan in $\Pi_{w}$.

The LP includes the following variables: For every
$$\pi =\langle a_{1},\ldots ,a_{k}\rangle \in \left(\Pi_{w}\cup \pi_{s}\right),$$
for t=1,…,k, we add a variable $X_{a_{t},t}$ (without repetition) representing the cost for the worker for an action *a* at time *t*. In addition, for every variable $X_{a_{t},t}$, we add a variable $Y_{a_{t},t}$, representing the supervisor cost for modifying the worker cost for the action associated with $X_{a_{t},t}$. The LP contains the following constraints:Action constraints: For every action, the modified cost should be greater or equal to the initial action cost
(2)$$\forall a_{t},t\in \Pi_{w},X_{a_{t},t}\geq C_{0}\left(a_{t},t\right).$$Plan constraints: The cost of every worker plan should be strictly greater than $C^{*}$.
(3)$$\forall \pi_{i}=\langle a_{1},\cdots ,a_{k}\rangle \in \Pi_{w},\sum_{t=1..k}X_{a_{t},t}>C^{*}.$$Supervisor plan constraints: $\pi_{s}$ is an optimal plan for jointly achieving the worker and the supervisor goals. For optimal plans, the cost of actions in $\pi_{s}=\langle a_{1},\cdots ,a_{m}\rangle$ should not be increased.
(4)$$\forall a_{t}\in \pi_{s},X_{a_{t},t}=C_{0}\left(a_{t},t\right).$$Supervisor modification cost constraints: The supervisor cost for modifying worker action costs is linear in the magnitude of the modification.
(5)$$\forall a_{t},t\in \Pi_{w},Y_{a_{t},t}=C_{s}\left(a_{t},t\right)\cdot \left(X_{a_{t},t}-C_{0}\left(a_{t},t\right)\right).$$

The objective function of $LP(\Pi_{w},C^{*},\pi_{s},C,C_{s})$ is to minimize the sum of the supervisor cost modifications over all variables.
(6)$$Min\sum_{a_{t},t}Y_{a_{t},t}.$$

Solving the LP, we obtain the revised costs for all actions that appear in plans in $\Pi_{w}$. As such, all plans in $\Pi_{w}$ have a cost higher than $\pi_{s}$, and the worker would choose $\pi_{s}$. Choosing different supervisor plans in $\Pi_{s}$ may result in different cost modifications. In particular, some plans in $\Pi_{s}$ require less cost modifications than others. Hence, we must solve $LP(\Pi_{w},C^{*},\pi_{s},C,C_{s})$ for every $\pi_{s}\in Pi_{s}$, and choose the minimal cost solution.

The CFM approach is shown in Algorithm 1. We create an LP for each $\pi_{s}\in \Pi_{s}$ and return the worker cost function with minimal supervisor cost. This algorithm necessarily returns an OSCF solution.
**Algorithm 1:** Cost Function Modification (CFM)
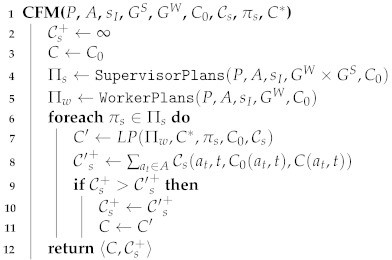


### 5.2. ICFM: Incremental Approach

The brute-force CFM approach requires as input all possible plans in $\Pi_{w}$ and $\Pi_{s}$. However, the size of each set can be exponential in the number of worker actions. As such, this approach does not scale. In this section, we provide an iterative version of our approach, called ICFM. The approach alternates between (1) computing a worker plan for a given cost function, incrementally adding constraints associated with the resulting plan to the LP, and (2) solving the linear program to obtain a new cost function. ICFM terminates when a worker plan is found that satisfies the supervisor goals. This approach does not guarantee an OSCF solution because we focus on a single $\pi_{s}$.

There can be several worker plans with cost $C^{*}$, and WorkerPlan may return $\pi_{s}$ prior to some $\pi_{w}\in \Pi_{w}$ that has a cost of $C^{*}$. To address this, we disqualify all such plans from the worker’s consideration by increasing the cost of plans that achieve the goals of both participants to be $C^{*}+\epsilon$. We do so by defining a slightly different planning problem, WorkerPlan-ϵ, adding a new proposition done which is the new goal of the worker. We add two actions that provide done, $achieved_{w}$, and $achived_{sw}$. $achieved_{w}$ has a preconditions $G^{W}$ and $\neg G^{S}$ and a cost of 0. $achieved_{sw}$, on the other hand, requires both $G^{S}$ and $G^{W}$ as preconditions, and has a cost of ϵ. In the revised problem the cost is increased to $C^{*}+\epsilon$. Hence, all worker plans of cost $C^{*}$ or less must be returned by WorkerPlan-ϵ before an SW-optimal plan is found. In the LP we omit the additional actions.

This ICFM approach is shown in Algorithm 2. We maintain an incrementally growing set $\Pi_{i}$ of worker plans, initialized to *∅*. The algorithm receives as input an SW-optimal plan $\pi_{s}$ and the associated worker cost $C^{*}$. In line 9, we solve $LP(\Pi_{i},C^{*}+\epsilon ,\pi_{s},C_{0},C_{s})$ to obtain the modified cost function for the worker under the constraints $\Pi_{i}$. In line 10, we call WorkerPlan-ϵ on the modified cost function to obtain a new worker plan $\pi_{i}$. This process continues until WorkerPlan-ϵ returns $\pi_{s}$.
**Algorithm 2:** Incremental CFM (ICFM)
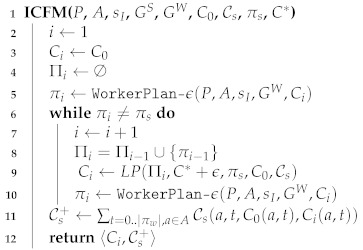


In the worst case, the approach will need to traverse all worker plans in $\Pi_{w}$. However, as each cost modification increases the cost of many possible plans, in practice, we may obtain sufficient cost increases for WorkerPlan to choose $\pi_{s}$ with orders of magnitude less constraints than the complete approach.

The solution to the final $LP(\Pi_{i},C^{*}+\epsilon ,\pi_{s},C_{0},C_{s})$ must have identical cost to the complete $LP(\Pi_{w},C^{*},\pi_{s},C_{0},C_{s})$. This is because $LP(\Pi_{i},C^{*}+\epsilon ,\pi_{s},C_{0},C_{s})$ is a roof [26] of $LP(\Pi_{w},C^{*}+\epsilon ,\pi_{s},C_{0},C_{s})$, and $C_{s}^{+}$ under the solution of $LP(\Pi_{i},C^{*}+\epsilon ,\pi_{s},C_{0},C_{s})$, must be a lower bound (as this is a minimization task) on the $C_{s}^{+}$ under the solution of $LP(\Pi_{w},C^{*}+\epsilon ,\pi_{s},C_{0},C_{s})$.

From the supervisor perspective, however, it may be that selecting a different $\pi_{s}$ would result in fewer cost modifications. Thus, ICFM is not guaranteed to return an OSCF.

However, we can still show that the solution is optimal for a specific SW-optimal plan. To this end, we make the following definition.

**Definition** **4.**
*A $\pi_{s}$-optimal solution for SWOPP is a pair $\left(\pi_{s},C^{\prime }\right)$ such that $\pi_{s}$ is an SW-optimal plan for the initial cost function, and $C^{\prime }$ is a revised worker cost function such that 1) any plan $\pi^{\prime }$ that is optimal for the worker (minimizes C(π)) is SW-optimal for $C^{\prime }$; and 2) $C^{\prime }$ minimizes the supervisor cost $C_{s}^{+}$ given $\pi_{s}$.*


The following section will show that ICFM returns a $\pi_{s}$-optimal solution.

#### Running Example

We now demonstrate ICFM over our running example. The input given to ICFM is $\pi_{s}=\langle move\left(n_{0},n_{3}\right),$$move\left(n_{3},n_{4}\right),move\left(n_{4},n_{2}\right),move\left(n_{2},n_{4}\right),move\left(n_{4},n_{g}\right)\rangle$. The cost of $\pi_{s}$ denoted $c^{*}=9$. The worker initial plan, computed in line 5, when i=1 is $\pi_{1}=\langle move\left(n_{0},n_{2}\right),move\left(n_{2},n_{4}\right),move\left(n_{4},n_{g}\right)\rangle$. The set of worker plans $\Pi_{1}$ now contains $\pi_{1}$ (line 8). An LP problem is created in line 9 with the plan constraint $X_{move(n_{0},n_{2}),0}+X_{move(n_{2},n_{4}),1}+X_{move(n_{4},n_{g}),2}>c^{*}+\epsilon$.

The solution to the LP problem is a revised cost function $C_{1}$ that modifies one action cost: $C_{1}\left(move\left(n_{2},n_{4}\right),1\right)=8$. After replanning (line 10), we get a new worker plan $\pi_{2}=\langle move\left(n_{0},n_{2}\right),move\left(n_{2},n_{0}\right),move\left(n_{0},n_{2}\right),move\left(n_{2},n_{4}\right),move\left(n_{4},n_{g}\right)\rangle$. Since $\pi_{2}$ is not $\pi_{s}$, we keep iterating (line 6). We define $\Pi_{2}=\Pi_{1}\cup \pi_{2}$. After solving the relevant LP, we get a revised cost function $C_{2}$ which modified the following action costs: $C_{2}\left(move\left(n_{0},n_{2}\right),0\right)=6$ and $C_{2}\left(move\left(n_{2},n_{4}\right),1\right)=3$. After another replanning, we get a new worker plan $\pi_{3}=\langle move\left(n_{0},n_{1}\right),move\left(n_{1},n_{g}\right)\rangle$. We define $\Pi_{3}=\Pi_{2}\cup \pi_{3}$. The relevant LP problem results in a new cost function that modifies the following action costs: $C_{3}\left(move\left(n_{0},n_{2}\right),0\right)=6$, $C_{3}\left(move\left(n_{2},n_{4}\right),1\right)=3$, and $C_{3}\left(move\left(n_{1},n_{g}\right),1\right)=4$.

This process continues five iteration until $\pi_{5}$ = $\pi_{s}$. Thus ICFM terminates and returns the worker cost function $C_{4}$ and the increase to the supervisor cost $C_{s}^{+}=\left(6-1\right)+\left(4-1\right)+\left(4-1\right)=11$.

## 6. Properties of ICFM

We now briefly discuss the properties of ICFM—termination, optimality, soundness, and completeness. This analysis applies only for a rational (optimal) worker. We also assume that a SW-optimal $\pi_{s}$ exists; otherwise, there is no valid solution to SWOPP.

ICFM terminates after a finite number of steps.ICFM is sound—it returns a cost function $C^{\prime }$, such that optimal planning under $C^{\prime }$ yields an SW-optimal plan.ICFM is complete with respect to an SW-optimal solution, $\pi_{s}$—if there is a cost function that forces the optimal planner to follow $\pi_{s}$, ICFM will find it.The ICFM solution is $\pi_{s}$-optimal.The SW-optimal plan $\pi_{w}$ for the worker does not incur additional cost for the worker ($C\left(\pi_{w}\right)=C_{0}\left(\pi_{w}\right)$).

**Lemma** **1.**
*Each plan $\pi \in \Pi_{w}$ can be computed by WorkerPlan at most, once during Algorithm 2.*


**Proof.** Assume (for contradiction) that the same plan was computed by WorkerPlan in two different iterations i,j, where i<j, that is $\pi_{i}=\pi_{j}$. When ICFM computes plan $\pi_{i}$ at step *i*, a plan constraint was added to the LP problem requiring that $C_{i+1}\left(\pi \right)>C^{*}$. This constraint holds at step j−1. Hence, $C_{j}\left(\pi_{i}\right)$ must also be greater than $C^{*}$. Given the supervisor’s plan constraints in the LP problem, $C_{j}\left(\pi_{s}\right)=C_{0}\left(\pi_{s}\right)$. Hence, $\pi_{s}$ must be computed by WorkerPlan before $\pi_{j}$ and the algorithm will terminate.    □

**Theorem** **1.**
*ICFM terminates after a finite number of steps.*


**Proof.** The ICFM loop (line 6) ends when $\pi_{i}=\pi_{s}$ which occurs when $C\left(\pi_{s}\right)\geq C^{*}$. By Lemma 1 each plan can be chosen by ICFM, at most, once. There is a finite number of possible plans of cost less or equal to $C^{*}$, and at least one plan $\pi_{s}$ with cost $C^{*}$. Hence, after a finite number of steps, the loop terminates.    □

**Theorem** **2.**
*ICFM is sound—it returns a cost function $C^{\prime }$, such that optimal planning under $C^{\prime }$ yields an SW-optimal plan.*


**Proof.** The resulted LP cost function $C_{i}$ is validated by the optimal classical planner (line 10). When ICFM terminates, $\pi_{i}=\pi_{s}$, validating that WorkerPlan under $C_{i}$ returns $\pi_{s}$. Hence, if this black box planner is sound, ICFM is also sound.    □

**Theorem** **3.**
*ICFM is complete with respect to an SW-optimal solution, $\pi_{s}$. That is, if there exists a cost function that forces the optimal planner to follow $\pi_{s}$, ICFM will find one.*


**Proof.** ICFM terminates only when $\pi_{i}=\pi_{s}$. ICFM always terminates; hence, ICFM returns a cost function that forces the optimal planner to follow $\pi_{s}$.    □

**Theorem** **4.**
*ICFM returns a $\pi_{s}$-optimal solution*


**Proof.** To show that ICFM returns a $\pi_{s}$-optimal solution, we need to show that, under the computed cost function $C^{\prime }$: (1) any plan that is optimal for the worker is SW-optimal under $C^{\prime }$ (Theorem 2). (2) $C^{\prime }$ minimizes the supervisor’s cost $C_{s}^{+}$ given $\pi_{s}$.To prove (2), we note that a solution for $LP(\Pi_{w},C^{*},\pi_{s},C_{0},C_{s})$, where $\Pi_{w}$ is the set of all possible worker plans with a cost less or equal to $C^{*}$, necessarily minimizes the supervisor cost given $\pi_{s}$. We denote the supervisor minimal cost as $C_{s}^{+*}$.We need to show that the optimal solution of the LP problem for a subset of worker plans $\Pi_{i}\in \Pi_{w}$, $C_{s}^{+}$, is smaller than $C_{s}^{+*}$. $\Pi_{i}$ contains worker plans with a cost less or equal to $C^{*}$, and as such, $\Pi_{i}\subset \Pi_{w}$. Each plan $\pi_{i}$ (line 10) adds a set of variables to the LP, and a set of constraints. Therefore, $LP(\Pi_{i},C^{*},\pi_{s},C_{0},C_{s})$ contains a subset of the variables and constraints in $LP(\Pi_{w},C^{*},\pi_{s},C_{0},C_{s})$. As the rest of the parameters to the linear program construction are identical, there are no additional variables and constraints in $LP(\Pi_{i},C^{*},\pi_{s},C_{0},C_{s})$ that do not appear in $LP(\Pi_{w},C^{*},\pi_{s},C_{0},C_{s})$As such, $LP(\Pi_{i},C^{*},\pi_{s},C_{0},C_{s})$ is a roof [26] of $LP(\Pi_{w},C^{*},\pi_{s},C_{0},C_{s})$, and $C_{s}^{+}$ under the solution of $LP(\Pi_{i},C^{*},\pi_{s},C_{0},C_{s})$, must be a lower bound (as this is a minimization task) on the $C_{s}^{+}$ under the solution of $LP(\Pi_{w},C^{*},\pi_{s},C_{0},C_{s})$. Hence, the solution of ICFM is $\pi_{s}$-optimal solution.For the other direction, the solution of $LP(\Pi_{i},C^{*},\pi_{s},C_{0},C_{s})$ provides cost modifications such that after applying them, the optimal planner WorkerPlan cannot find a worker plan of cost less or equal to $C^{*}$. Therefore, the solution of $LP(\Pi_{i},C^{*},\pi_{s},C_{0},C_{s})$ also maintains the constraints in $LP(\Pi_{w},C^{*},\pi_{s},C_{0},C_{s})$.    □

## 7. Empirical Analysis

We now provide an empirical study of our algorithm on a set of modified benchmarks. We describe the domains and how we modified different features to control for the complexity of solving the SWOPP problem. To compute optimal worker plans, we use the fast-downward (FD) planner [25], running an $A^{*}$ variant with the LMCut heuristic [27]. To obtain the sets of all plans $\Pi_{s}\;\mathrm{and}\;\Pi_{w}$ of cost less than $C^{*}$, we used BFS. The experiments were run on an Intel i7-8550u CPU at 1.99 GHz with 16 GB of RAM.

We adapt the following domains from the classical planning competition (IPC) [28]. In all domains, we allow the supervisor to increase the cost of any action to any natural number, and the cost to the supervisor is the difference between the original cost and the increased cost.

Grid: The grid domain is an n×m fully connected map where the worker’s task is to move from one location to another. We assign a difficulty level to each passage between neighboring cells, and the action costs represent this difficulty. The worker’s goal is to visit a set of cells, while the supervisor’s goal is for the worker to visit additional cells.

Blocks: The well known blocks world describes a domain where stackable blocks need to be re-assembled on a table using a robotic arm. The goals of the worker and the supervisor are to achieve different block arrangements. The supervisor’s goal is to achieve a specific block arrangement at least once during the plan execution (we do not require this arrangement to hold at the end of the execution).

Logistics: The task is to transport packages within and between cities using trucks and airplanes. The worker’s goal is to deliver a subset of the packages, while the goal of the supervisor is for the worker to deliver additional packages.

### 7.1. Varying the Domain Difficulty

To better understand the behavior of ICFM, we vary the domains according to several factors, which are intuitively related to the difficulty of solving the problem.

First, we look at the effort required for the worker agent to achieve the goals of the supervisor on top of its own goals. We hypothesize that, when more effort is required from the worker agent, there would be significantly more disagreements between the worker and supervisor plans. Hence, the supervisor must impose more modifications on the worker’s cost function, resulting in higher costs for the supervisor.

For example, in the grid domains, the cells that the supervisor requires the worker to visit may be far from the goal cells of the worker, requiring larger detours by the worker. Analogously, in education, when the teacher needs the student to master multiplication of large numbers, and the student has already mastered multiplication of small numbers, then we can argue that the required effort from the student is relatively small. On the other hand, if the student has only mastered addition, then a higher degree of effort is needed by the student before accomplishing the goals of the teacher.

### 7.2. Baseline Algorithm

We compare ICFM to a baseline algorithm, which increases the costs of all currently available worker actions by a sufficiently high amount, so that they will not be selected by the worker’s plan. The algorithm begins with computing $\pi_{s}$, and then starts simulating the execution of $\pi_{s}$. Following the execution of each action $\pi_{s}\left(t\right)$, for every action $a^{\prime }\neq \pi_{s}\left(t\right)$ whose preconditions are met, we increase the cost of $a^{\prime }$ by $C(\pi_{s}\left(t+1,\ldots ,\right|\pi_{s}\left|\right)+\epsilon$ so that it will not be selected by the worker. Similar to ICFM, this algorithm causes the worker to eventually choose the optimal plan with respect to $C_{0}$. This baseline algorithm is very fast compared to ICFM because it avoids the multiple calls to the optimal planner, but it increases the cost of many actions that would not be chosen by the worker. Thus, we expect it to incur a much higher cost for the supervisor.

### 7.3. Results

Table 1 shows the results of running CFM and ICFM on the Grid, Blocks, and Logistics domains described above. The results focus on the time required to produce a solution, as well as the resulting supervisor cost, which corresponds to the efficiency of the solution.

In the table, *T* denotes the additional effort required to achieve the supervisor goals (L= low, H= high). Cost and length denote the original cost and plan length for the optimal plan achieving only the worker’s goals ($\pi_{w}$), and the optimal plan achieving the joint goals ($\pi_{s}$). For the three algorithms that we compare, the baseline algorithm, CFM and ICFM, *Time* denotes elapsed wall clock time. Lower time to termination is preferable. $C_{s}^{+}$ denotes the cost to supervisor, which is the sum of cost increases to $C_{0}$. We prefer a lower increase to $C_{0}$; hence, a lower $C_{s}^{+}$ is better.

$|\Pi_{s}\left|\;\mathrm{and}\;\right|\Pi_{w}|$ denote the amount of possible supervisor and worker plans smaller or equal to $C^{*}$, respectively. As CFM must iterate through all $\Pi_{w}$ before finding the optimal plan, this number correlates with the difficulty of the particular problem instance. $\pi_{i}$ denotes number of plans that were computed in ICFM before $\pi_{s}$. ICFM is an iterative algorithm, requiring a new plan to be computed at each iteration. Hence, $\pi_{i}$ is the number of iterations that ICFM required before identifying a good plan. $\pi_{i}$ is also the number of plans in $\Pi_{w}$ that ICFM considered. Recalling that CFM must check all plans in $\Pi_{w}$, $\pi_{i}$ is an indication of the efficiency of ICFM.

During the experiment, ϵ was set to 1, i.e., modifications to the cost function are in units of 1. The BFS component of CFM was terminated if it did not enumerate all needed plans within 10 min.

As shown in the table, for all domains, the cost to the supervisor $C_{s}^{+}$ is significantly lower for CFM and ICFM than the baseline but requires much more computation time. For many domains, such as logistics-2,2,6 (*H*), CFM and ICFM produce an order of magnitude improvement of supervisor’s cost over the baseline. For small instances, CFM returns an equal or better supervisor cost $C_{s}^{+}$ than ICFM, requiring less computation time. This is because ICFM must solve multiple LPs, and compute multiple plans. However, the computational time of CFM is growing with $|\Pi_{s}\left|\;\mathrm{and}\;\right|\Pi_{w}|$, making CFM infeasible for bigger instances, such as grid-3 × 8(*L*). When the additional effort for achieving the supervisor goals is high (T=H), ICFM requires more replanning and constraints than when additional effort is low (T=L).

## 8. Applying SWOPP in the Classroom

We now describe a user study applying SWOPP to an e-learning application in which students solve math problems. The setting includes a teacher, a student, a set of math skills, a set of questions such that each question is associated with a skill, and a partial ordering over the skills representing the set of prerequisite skills that are necessary for acquiring a new skill. A student acquires a skill after successfully solving a set of problems associated with the skill. Our goal is to show that, although students can choose which math questions to solve, SWOPP directs students to solve math skills that their teacher assigns them. Thus, student practice math skills that they need to improve, even though they can choose to practice other skills.

The user study was run in collaboration with the teachers and school management towards the end of the academic year. We received access to the students’ grades in the various math topics. The study was conducted in the school’s computer lab during regular school hours, under the teacher’s supervision.

### 8.1. E-Learning Game

We created an e-learning game framed as an amusement park in which the goal is to reach a castle by traversing paths that include carnival rides and gates (Figure 2) (A demo of this game can be seen at https://www.youtube.com/watch?v=GbaotmVRsOA.). Each carnival ride represents a skill. To acquire the skill, the student needs to open the gates leading to the ride. There is a repository of questions from a specific skill for each gate. When failing to answer a question correctly, the student can skip to an alternative question from the same skill. A gate is opened only after correctly solving a predetermined number of questions associated with the skill. A gate can be accessed by the student only if the student has the relevant prerequisite skills.

Each student receives a personally tailored map, consisting of 8 possible paths leading from the entrance of the park to the castle. The gates along each of the paths in the map contain questions belonging to 3–5 skills. Paths differ in the skills that they practice for the student, and the number of questions to solve (gates to open). Students can reach the castle by traversing any path and opening all of the gates in the path. An example map from the study is shown in Figure 2. The student avatar (a bull) is at the starting point. The leftmost path includes 4 skills in the core topic of geometry, such as computing areas of various shapes, and properties of quadrilaterals.

We created an SWOPP formalization for this e-learning game, providing a personalized map for each student. The map included multiple paths for core topics that students cam practice. For example, when a student practices geometry, there are two possible paths: one path practicing quadrilateral properties, and another path practicing areas and perimeters of polygons. In each path, different rides represent the skills the student need to practice in that topic. Skills must be practiced in the correct order, such students first practice perquisites to later skills. For each skill, there is at least one gate that the student must open to progress in the path. To open the gate, the student needs to practice the skill and solve a predetermined number of questions on that skill. When the gate is open, the student can continue to the next gate in the path. When practicing the last skill in the path, the student reaches the castle and the game is over. Figure 3 shows this game flow.

### 8.2. Mapping the e-Learning Game to SWOPP

Mapping this game to SWOPP, each skill is represented as a node. We set the edges between the nodes (skills) to represent perquisites, as in the map. We define an action solve(s) for each skill *s* in the map. The cost of solve(s) is personalized given the student’s preferences over which math skills she prefers to practice.

The SWOPP worker is the student, and the SWOPP supervisor is the teacher. The goal of the student (worker) is to reach the castle node. The teacher’s (supervisor) goal is for the student to acquire a specific needed skill, i.e., to reach a specific node. The worker cost function represents the personalized effort required for that particular student for solving the questions associated with the gate leading to that skill. We set the supervisor cost function to be proportional to the increase in the student cost function, representing the decrease in perceived student freedom to choose as more questions are added. We decided not to use a time-based cost function because we suspected that students would find it confusing that gates appear and disappear while playing.

### 8.3. Construction of the Personalized Cost Function

Eliciting an appropriate cost function for humans is a difficult task [29]. In this study, as a first step, we create the cost function based on the student’s stated pairwise preferences, e.g., Reference [30]. In many cases, the student’s preferences is in correlation with the student’s performance (grades). However, as we have seen in our data, this is not always the case. Hence, we choose to use a subjective self-reporting method instead of using the students grades. Of course, in some cases, the student’s perception of their preferences is inaccurate. More complex elicitation methods may produce more accurate cost functions. For example, if we were running a game over a longer period of time, including multiple sessions, we could perhaps use the earlier choices that student make within the game to elicit their preferences, and use this information to shape their personalized experience later on. We leave such more complex elicitation methods to future research.

The cost function for each student is based on a pairwise preference questionnaire, conducted a week before the study. In the questionnaire, we presented 3 different choices between 2 questions to solve, each question from a different skill. We emphasized to the students not to solve the questions, only to tell us which question they preferred to solve. Table 2 shows an example of a student response.

Using the stated pairwise preferences, we create a personalized cost function for each student, that assigns higher costs for questions in skills that are less preferred by the student. Specifically, the student’s cost function set a cost for solving 5 questions from a given skill (opening a gate in the game). As shown in Table 2, each student had to specify preferences over 3 pairwise questions, for the following pairs *f-nn*, *g-nn*, *g-f*, where *f* denotes fractions, *g* denotes geometry, and nn denotes natural numbers. Let $p_{s_{1},s_{2}}$ be the preference on a scaled of 1 to 5 for the pairwise preference between skills $s_{1}$ and $s_{2}$. We compute the following personalized cost for each skill:$$c_{f}=p_{f,nn}+\left(6-p_{g,f}\right),$$
$$c_{g}=p_{g,nn}+p_{g,f},$$
$$c_{nn}=\left(6-p_{f,nn}\right)+\left(6-p_{g,nn}\right).$$

For example, in Table 2, we have that $p_{f,nn}=1$, $p_{g,nn}=5$, $p_{g,f}=4$. Hence: $c_{f}=3,c_{g}=9,c_{nn}=6$.

We set the teacher’s goal to be the skill where the student received the lowest grade during the academic year. We increase the cost of the skill that was chosen to be the teacher’s goal by 2. This is done to mitigate the subjectivness of the student’s cost function. In particular, students often overestimate their ability to handle certain skills, compared to the teacher’s grades [31].

Once we set the student personalized cost function, we solve SWOPP using the ICFM algorithm. The solution to the SWOPP problem is a personalized cost function for the student that considers the goals of the student, to reach the castle, as well as the teacher’s goal, for the student to acquire a specific skill. The resulting cost function induces a personalized map for the student by specifying the number of gates required for reaching each skill. The added gates are deduced from the increase in the edge cost in the SWOPP solution. That is, the higher SWOPP increased the original cost of traversing an edge, the more gates where added on the path corresponding to that edge.

### 8.4. Elements of Gamification

Our e-learning application above is a type of serious game, where the goal is to improve the learning experience [32]. Many other serious games were suggested in education [33]. The main difference between previous e-learning games and ours is in focusing on guiding students towards skills they need to practice, while maintaining the the freedom to choose their activities. We also allow for personalization, providing a particular game map based on the specific student’s preferences and difficulties.

To maintain interest in the game, we use several well known elements of gamifications [34]:Levels/Milestones: Each gate on the map represents a milestone. When the student answers correctly the related questions to the gate, the gate is opened and the milestone was achieved. In addition, each carnival ride is also a milestone that can be achieved after opening all the gates in the path. The gates and rides are originally grayed out and become colored only after the milestone was achieved.Points: When enabling a carnival ride, the student receives a point. Points are shown on the top left corner of the map.Avatar: In the beginning of the game, each student can choose an avatar from 6 different choices: a human student, and 5 animals—a giraffe, an elephant, a tiger, a bull, or a deer.Visual elements: We use many visual elements to represent concepts in the e-learning domain. Carnival rides represent skills, and each ride is accompanied by a flag providing a visual cue for the skill. Gates represent questions. Colored rides or gates represent already achieved skills, and so forth.

### 8.5. User Study

The study included 75 K5 students from three heterogeneous classes in a public elementary school. The study was conducted during regular school hours with the teachers support. IRB approval was obtained to run the study. When asked to rank their familiarity with math e-learning software (Figure 4), 35% of the students said that they were not familiar at all with e-learning software (score 1). Only 16% of the students reported they were familiar e-learning (score 4–5). When asked what is preferable, traditional learning from a book or using an e-learning software (Figure 5), only 13% reported that a book was preferable (a score of 1–3), while 87% reported that e-learning was more preferable (score 4–5).

The study included questions from 20 different skills, spanning three core topics from the K5 curriculum: fractions, geometry, and natural numbers. We obtained prior grades (for each skill) for the students from the school, independent of the experiment. For each student, we set the teacher goal to practice the skill associated with the lowest grade for the student.

We designed a between-subject study by randomly assigning students to one of three conditions: First, students who receive a default map with no supervisor intervention (*D*), which included a single gate for each of the skills. Second, students who receive a personalized SWOPP map (SWOPP), that varies the number of gates to open for different skills. Third, students who receive a no-choice map (NC) in which only the teacher’s goal path is active. In contrast, for both *D* and SWOPP, students were able to freely choose any path in the game.

For SWOPP, the student goal was to reach the castle; the supervisor goal was for the student to acquire skills in the weakest skill for the student, as determined by the teacher’s grades. A personal map for each student was generated, following our approach, by increasing the cost (adding gates to paths) to some of the skills. All students received a brief explanation about the e-learning software and were told to reach the castle. The study was limited to 45 min. A post-study user experience survey was conducted.

### 8.6. Experiment Protocol

We obtained from the teachers all students’ grades for the various geometry skills in the study. A week before the study, all students filled the pairwise preference elicitation questionnaire, and were given a consent form to be filled and signed by their parents.

Given the pairwise preferences and the teacher’s grades, we construct for each student a personalized cost function. We define an SWOPP problem and solve it for each student, as explained above. We construct a personalized map for each student using the solution of the SWOPP problem. We also split the students randomly to the three conditions (*D*, NC, SWOPP).

The students played the e-learning game in the school’s computer lab. As the lab did not have enough computers for all students, when a student finished the study, another student took her place. Students entered at an arbitrary order, defined by the teachers.

Each student received a personal code and logged into the system to receive their personalized map. The student entered the game and chose an avatar.The student then began the game, receiving the game map matching to the specific condition and personal goal and cost function.

The students played the game without any time limitation until they reached the castle. Nine students decided in the middle of the game that they do not wish to continue. After finishing the game (whether successfully or not), the students were asked to fill a post-study questionnaire. After filling the questionnaire, the students were rewarded with an ice cream.

### 8.7. Study Results

Table 3 lists the number of students in each condition, and the number of times students in each group reached the teacher’s goal ($G^{S}$, practicing their weakest skill) and the student’s goal ($G^{W}$, getting to the castle). As shown in the table, 65% of students using the SWOPP map achieved the teacher’s goal, while only 17% of the students using the *D* map achieved the teacher’s goal ($\chi^{2}\left(1,N=45\right)=13.612,p<0.0005$). Unsurprisingly, all of the students who finished the game in the NC (no choice) condition achieved the teacher’s goal. The number of students who chose to quit and did not reach their goal (3) was the same in all conditions.

In the post-study survey, we asked students to self report on how much choice they felt they had in the game (Likert scale from 1 to 5). The average perceived level of choice in the SWOPP condition, 4.08, was significantly higher than that in the NC condition, 1.62. ($\chi^{2}\left(1,N=41\right)=20.2567,p<0.0005$). The average reported perceived level of choice in the *D* condition, 3.66, was also significantly higher than that in the NC condition ($\chi^{2}\left(1,N=44\right)=15.4795,p<0.0005$). There was no statistical significant difference between the reported choice level in the SWOPP and *D* conditions ($\chi^{2}\left(1,N=41\right)=0.7738,p<0.1$). We can conclude that students in SWOPP and *D* reported that their choice of questions was unconstrained, while users in the NC did not. Participants in all conditions reported that they enjoyed playing the game.

Table 4 analyzes task performance in the three different conditions. As shown in the table, the number of total questions solved in average in the SWOPP (50.09) and *D* conditions (60.25) was higher than that of the NC condition (40.9). That is, students with a higher level of choice solved more questions. The average portion of correct answers in the SWOPP condition (40%) was higher than that of the NC condition (35%), and students solved more questions in SWOPP than in NC (50.09 vs. 40.9). This demonstrates that using SWOPP allowed students to practice their necessary skill set, as determined by the teacher, without deterring their perceived level of choice or their motivation.

To summarize, our user study shows that SWOPP can be used in real world problems. We were able to formalize the e-learning task as an SWOPP instance and use our algorithms to rapidly produce cost functions and, hence, personalized maps, in a real scenario. Most students who used received the SWOPP map, although they could choose which path to follow, completed the teacher’s goals. This shows that SWOPP indeed guides students to practice their assigned skills, even though they could have chosen to practice other skills.

These results clearly indicate the validity of the SWOPP model, and the usability of the ICFM algorithm, for tackling real world problems.

### 8.8. Case Study

Finally, we review a specific case study of a fifth grade student, who was assigned randomly to the SWOPP condition. The grades (strengths) of the student obtained from the teacher were 100 in fractions, 81 in geometry, and 100 in natural numbers. According to the student’s grades, the student masters fractions and natural numbers computations but needs to practice geometry skills and, more specifically, to practice the properties of squares (leftmost path in Figure 2).

The student ranked the skills by preference (Table 2). The supervisor goal is set to practice the least preferred skill, that of quadrilateral properties in geometry. We compiled the student’s preferences into a cost function, assuming that questions associated with preferred skills require less effort. The cost to open a gate with a question practicing the natural number skill was set to 6, the cost to open a gate practicing the fraction skill was set to 3, and the cost of opening a gate practicing the geometry skill was set to 11.

Figure 2 shows the SWOPP tailored map for this student. For example, the algorithm increased the cost of the rightmost path by adding two additional gates to this path. The algorithm did not add any gates to the intended path (leftmost path). The student chose the intended path (leftmost path) and practiced the quadrilateral properties skill as required by the teacher. In the post-study survey, this student reported a high perceived level of choice (4 out of 5). When asked about playing the game, this student reported counting the number of gates in each path and following the path with the minimal amount of gates.

## 9. Conclusions and Future Work

We present the supervisor-worker problem where a supervisor must guide a worker towards achieving the supervisor’s goals, while allowing the worker to choose between several possible actions. We suggest that this approach is appropriate for the educational domain, where it is preferable for a teacher to allow a student to choose, while ensuring that the student will learn the necessary skills. We provide a formal definition for this setting, and then suggest an algorithm, ICFM, which is sound and complete, and optimal from the worker’s perspective. We apply our approach to a real world problem of generating math questions to students, showing we can guides students to practice skills that are challenging for them, while maintaining the perception of choice and motivation. The user study demonstrates the ability of SWOPP to personalize learning paths for students in a way that allows them to learn necessary skills without deterring their motivation. For teachers, this is a way to easily define gaming environments for their students to allow them to learn necessary skills, all that is needed is to provide their preferences over skills, which can be inferred from their performance in class.

Hence, the contributions of this work are:We define the supervisor-worker setting and show its applicability interesting real world problems. The e-learning game that we developed attends to the usefulness of SWOPP in real world applications.We suggest an optimal algorithm for solving SWOPP problems (CFM), and an approximate efficient algorithm (ICFM), that is able to solve real world problems. The empirical results over artificial problems show the power of ICFM to scale up, while maintaining a reasonable cost for the supervisor. The formalization of the e-learning problem as SWOPP shows that ICFM can solve real world problems.We demonstrate the usefulness of SWOPP in an e-learning game. Conducting a user study, we evaluated the game with K5 children in a public elementary school. This shows that SWOPP can be used in a real world application, providing value to people.

A limitation of our user study is that we had limited ability to conduct thorough surveys with the K5 students, uncovering, e.g, deeper insights about their view of the choices given to them. Conducting such surveys is left for future work. One limitation of the SWOPP model requires a worker cost function which may be hard to obtain. Our method for obtaining the student cost function using preferences reported by students is, at best, crude. An obvious extension of this work is to start without knowing the student cost function, learn from the student choices their true cost function, and replan to revise costs accordingly.

There are many possible extensions to SWOPP: First, an interesting extension is to adapt the system behavior online by, e.g., analyzing the time spent on earlier questions, as in some advanced intelligent tutoring systems [35]. Second, we choose a fixed number of questions to be a proof of knowledge of a skill. An interesting future direction can be to use a better adaptive criterion. For example, we could have allowed students to solve only a few difficult questions to open a gate, with a fall back to additional easier questions should they fail. Third, it can be interesting to investigate different types of cost functions for the supervisor. For example, in our education settings, we assume that the teacher incurs a higher cost for larger modifications. Incurring a constant teacher cost for each modification is an interesting alternative to our teacher cost function. Finally, it can be interesting to study settings in which teams of workers work together; hence, a joint cost function is needed.

## Figures and Tables

**Figure 1 sensors-21-01965-f001:**
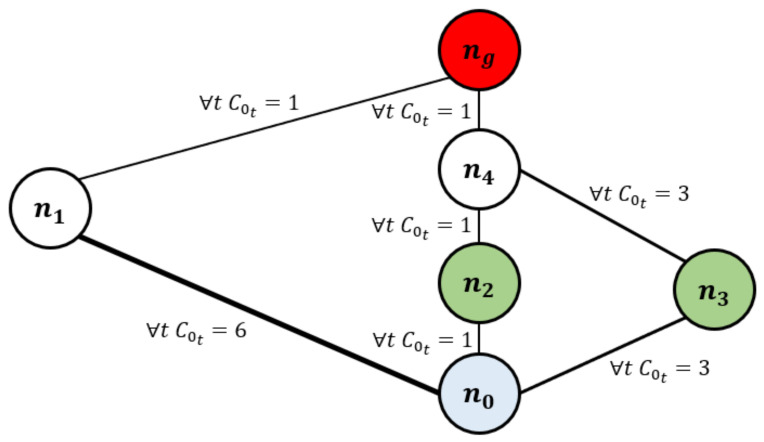
The navigation example.

**Figure 2 sensors-21-01965-f002:**
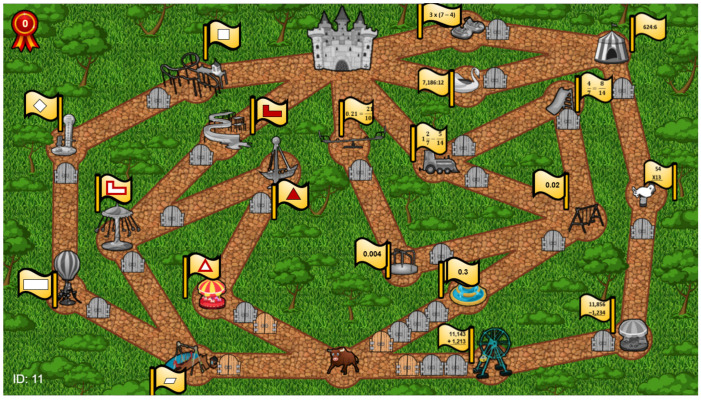
SWOPP (Supervisor-Worker Problem with Partially Overlapping goals)-induced map for carnival game for a given student in the study. The intended goal of the teacher for the student to practice the properties of quadrilaterals (leftmost path). The cost of the first skill in all other paths was increased by the algorithm.

**Figure 3 sensors-21-01965-f003:**
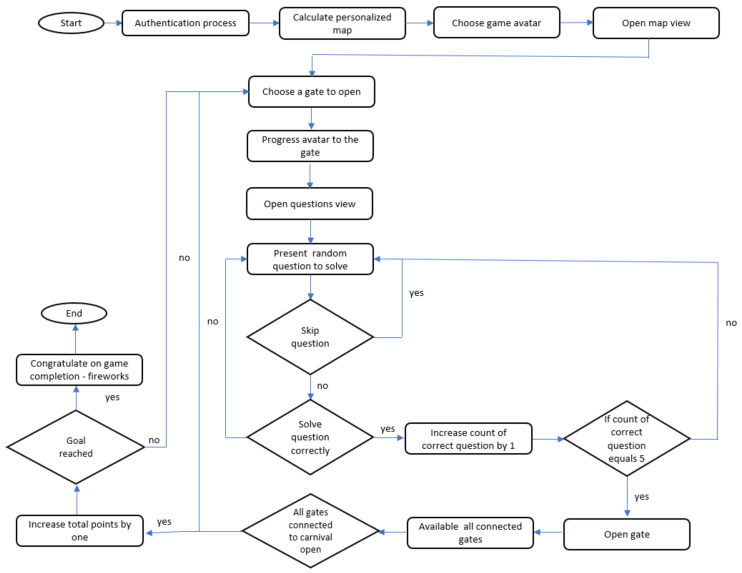
Schematic representation on the game flow.

**Figure 4 sensors-21-01965-f004:**
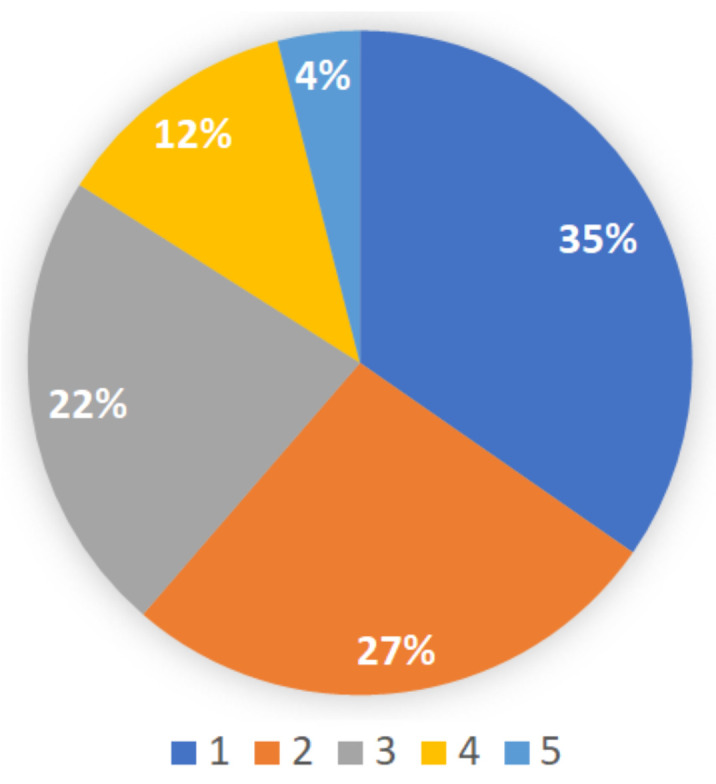
Previous familiarity with math e-learning software. Ranking from 1 (none) to 5 (experienced).

**Figure 5 sensors-21-01965-f005:**
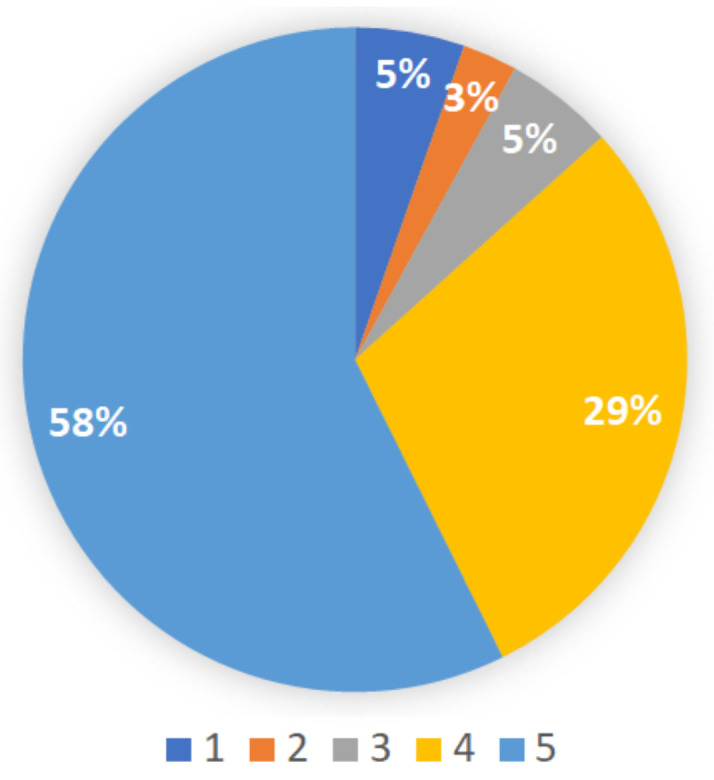
Student preferences over traditional learning using a book (1) or using an e-learning software (5).

**Table 1 sensors-21-01965-t001:** Baseline and Incremental Cost Function Modification (ICFM) performance for different domain difficulty (H,L), and costs and plan lengths for $\pi_{w}$ and $\pi_{s}$.

	Cost		Length		Baseline		CFM				ICFM		
Size (T)	$\pi_{w}$	$\pi_{s}$	$\pi_{w}$	$\pi_{s}$	Time	$C_{s}^{+}$	$|\Pi_{s}|$	$|\Pi_{w}|$	Time	$C_{s}^{+}$	Time	$C_{s}^{+}$	$\pi_{i}$
Grid													
3 × 4 (*L*)	1	14	1	3	**0.84**	52	1	6	1.96	**24**	4.31	30	5
3 × 4 (*H*)	1	25	1	9	**0.45**	215	8	253	7.44	**91**	13.87	**91**	19
3 × 4 (*L*)	13	16	5	5	**0.37**	79	2	5	4.3	**6**	2.67	12	3
3 × 4 (*H*)	13	18	5	7	**0.43**	124	2	18	1.57	**16**	4.83	22	6
3 × 6 (*L*)	11	20	7	9	**1.03**	212	2	505	36.76	**31**	24.51	46	24
3 × 6 (*H*)	11	22	7	11	**1.1**	255	2	1454	64.07	**42**	54.15	62	49
3 × 8 (*L*)	17	29	9	17	**1.4**	517	BFS timeout				86.34	**79**	51
3 × 8 (*H*)	17	38	9	15	**1.21**	642	BFS timeout				184.41	**253**	110
Blocks													
2,5 (*L*)	9	12	7	9	**1.07**	132	10	2	4.6	**4**	7.36	6	4
2,5 (*H*)	9	18	7	13	**2.25**	272	14	143	82.67	**27**	141.02	39	39
4,7 (*L*)	13	16	11	13	**6.59**	227	3	5	56.99	**6**	41.8	**6**	3
4,7 (*H*)	13	19	11	15	**9.23**	305	BFS timeout				184.35	**19**	17
Logistics													
2,2,6 (*L*)	4	6	3	5	**0.96**	59	5	4	3.24	**6**	7.08	**6**	4
2,2,6 (*H*)	4	15	6	10	**1.23**	322	5	51	26.71	**16**	16.62	**16**	9
3,3,9 (*L*)	15	19	4	8	**2.42**	430	BFS timeout				462.04	**39**	67
3,3,9 (*H*)	15	21	4	14	**9.54**	756	BFS timeout				591.31	**63**	86

**Table 2 sensors-21-01965-t002:** An example student’s preferences over skills.

Skill	Preference	Skill
	1 2 3 4 5	
Fractions	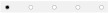	Natural numbers
Geometry	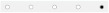	Natural numbers
Geometry	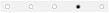	Fractions

**Table 3 sensors-21-01965-t003:** Goal performance (achieved $G^{S}$, not achieved $G^{S}$, and number of users in each condition).

Map	$G^{S}$	$\neg G^{S}$	$\neg G^{W}$	Participants
SWOPP	15	5	3	23
NC	21	0	3	24
*D*	5	20	3	28

**Table 4 sensors-21-01965-t004:** Question level performance.

Map	# Solved Questions	% Correct Answers	# Skipped Questions
SWOPP	50.09	40%	15.22
NC	40.9	35%	18.53
*D*	60.25	38%	15.23

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
