# Peer review of "Supervisor-Worker Problems with an Application in Education"

_sensors, 2021, doi:10.3390/s21061965_

Round 1
Reviewer 1 Report
The application of personalization of learning paths is a multifaceted topic and the work presented addresses it from a creative and original perspective.
The presentation of the problem and the context in which it has been investigated has been described in detail.
The rationale for the work is correct. Perhaps it would have improved the work to describe different approaches when dealing with this topic of personalization of learning paths. It is convenient to contextualize the framework on which we are basing ourselves. This would have enriched the presentation of the article and also the discussion of the results.
The description of the implementation of the experience is appreciated as it contributes to understanding the research.
The analysis of the data has been rigorous. Their representation in the tables has been adequate.
The results obtained are consistent with the approach. The conclusion integrates the results, but should detail and present in a disaggregated form the achievements in terms of the three contributions that the article proposed.
Author Response
Dear reviewer, we thank you for reviewing our paper. We appreciate the review and the suggestions for improvement. We have addressed the issues raised by you, and the modifications to our manuscript. For your convenience the changes in the manuscript from previous submission were highlighted in blue.
Comment 1
Describe different approaches when dealing with this topic of personalization of learning paths
Response
Thank you for this suggestion, we have revised the related work chapter to include a discussion of sequencing educational content, which we have expanded to include personalized learning following your suggestions.
_________________________________________________________________________
Comment 2
The conclusion should detail and present in a disaggregated form the achievements in terms of the three contributions that the article proposed
Response
We revised the conclusion to be clearer about the research contributions and achievements.
Reviewer 2 Report
I enjoy reading about use of novel methods in educational measurement, and reading about SWOPP for balancing the goals of the instructor and student was interesting and informative. The problem I see is that a majority of the paper is devoted to discussing SWOPP itself, and only about a third of the paper is devoted to the actual educational context. Further, the measurement methods and experimental design are both insufficient to draw trustworthy and robust conclusions about the effect of using SWOPP-based methods on student learning and affect.
Major Concerns Include:
The Review of Literature makes it sound like this is the first time SWOPP has been used in educational settings. Is this true? Much of the paper reads a lot like a book chapter. Some of the mathematical formalization may be warranted if this is the first time these methods have been used; otherwise, it might be better to include this material in a Supplement and instead focus on more conceptual explanation as well as the actual research context. In either case, more rigorous discussion of the application, and its impact on students, is needed to make a case that these methods work and are desirable.
Table 1 needs more detailed explanation regarding what it is supposed to show.
It’s not clear how the personalized map for each student was developed. What prior information was this based on, and how was the information used to generate the personalized map?
In the beginning of the study, the authors make the case that being given options increases interest and motivation. I am surprised the authors didn’t include a validated survey of interest or motivation with respect to the topics. They only asked about “perceived choice”, which isn’t very intellectually satisfying as it seems tautological that students who are given more choices will perceive having more choices.
I don’t buy into the argument that shorter response time is associated with greater engagement, nor significance of the idea that having more choice would facilitate shorter response times. This result isn’t surprising given the argument that students will tend to choose easier questions whereas the teacher will tend to assign harder questions—easier questions take less time to solve than harder questions. This doesn’t necessarily relate to engagement or learning at all.
Overall, I found the experimental design, measurement, and subsequent reflection and analysis to be pretty limited. The authors could have done more to think about the actual target constructs they intended to measure, and to utilize a few of the many validated surveys out there which measure these types of constructs.
I agree that calculating a cost function based on student self-reported preferences is a fairly crude way to go about this, especially if the goal is to develop a system that can be used with many students at a time without having to first collect preferences using a survey. A more scientific way to go about this would be to give the questions as a pilot test, and rank the questions by relative difficulty, or use previously piloted questions which have empirically-derived estimations of difficulty. The idea is that more advanced students will be more likely to answer more difficult questions correctly. Providing the pilot dataset is representative, the difficulty-based costs could be generalized across all students. Wright and Stone’s book Best Test Design addresses these types of problems.
Author Response
Dear reviewer, we thank you for reviewing our paper. We appreciate the detailed review and the many suggestions for improvement. We have addressed the issues raised by you, and the modifications to our manuscript. For your convenience the changes in the manuscript from previous submission were highlighted in blue.
Comment 1
The Review of Literature makes it sound like this is the first time SWOPP has been used in educational settings. Is this true? Much of the paper reads a lot like a book chapter. Some of the mathematical formalization may be warranted if this is the first time these methods have been used; otherwise, it might be better to include this material in a Supplement and instead focus on more conceptual explanation as well as the actual research context.
Response
Indeed, this is the first time SWOPP was formalized and has been used in educational settings. We have revised the abstract/introduction to be clearer about this.
________________________________________________________________
Comment 2
In either case, more rigorous discussion of the application, and its impact on students, is needed to make a case that these methods work and are desirable.
Response
Thanks for this suggestion. The success of our approach shows it is possible to personalize learning paths to students in a way that allows them to learn necessary skills without deterring their motivation.
For teachers, this is a way to easily define gaming environments for their students to allow them to learn necessary skills, all that is needed is to provide their preferences over skills, which can be inferred from their performance in class. We revised the discussion section to include the above.
________________________________________________________________
Comment 3:
Table 1 needs a more detailed explanation regarding what it is supposed to show.
Response:
This table shows the performance of our 2 suggested algorithms (CFM,ICFM) and a baseline algorithm. We tested 3 different popular domains from the planning community in different settings relevant to the SWOPP problem. We added more details about the table in section 7.3.
________________________________________________________________
Comment 4:
It’s not clear how the personalized map for each student was developed. What prior information was this based on, and how was the information used to generate the personalized map?
Response:
The personalized map was developed using the student individual cost function. The cost function for each student was based on a preference questionnaire, conducted prior to commencing the study. We added many details in section 8 describing our motivation and practice.
________________________________________________________________
Comment 5:
In the beginning of the study, the authors make the case that being given options increases interest and motivation. I am surprised the authors didn’t include a validated survey of interest or motivation with respect to the topics. They only asked about “perceived choice”, which isn’t very intellectually satisfying as it seems tautological that students who are given more choices will perceive having more choices.
Response:
The perceived choice question was not trivial. The alternative choices presented to students in the SWOPP condition required more effort from students compared to the intended path (that practices the teacher’s required skills). The fact that most students chose the intended path while still feeling they had choice speaks to the success of our approach. We agree with the reviewer that including a validated survey of interest of motivation with respect to topics would be better. However we had limited access to the students in the after school program. We did collect a preference questionnaire from students about different topics, and used it to construct the student cost function.
________________________________________________________________
Comment 6:
I don’t buy into the argument that shorter response time is associated with greater engagement, nor significance of the idea that having more choice would facilitate shorter response times. This result isn’t surprising given the argument that students will tend to choose easier questions whereas the teacher will tend to assign harder questions—easier questions take less time to solve than harder questions. This doesn’t necessarily relate to engagement or learning at all.
Response:
This is indeed debateable, and we remove this part from the result section.
________________________________________________________________
Comment 7:
Overall, I found the experimental design, measurement, and subsequent reflection and analysis to be pretty limited. The authors could have done more to think about the actual target constructs they intended to measure, and to utilize a few of the many validated surveys out there which measure these types of constructs.
Response:
Thanks for these suggestions. We emphasize that we evaluated SWOPP in a real classroom with real students, not subjects, hence we had limited control on the design of the testing environment. Despite these constraints, we were able to show the efficacy of SWOPP for guiding students’ towards acquiring needed concepts.
________________________________________________________________
Comment 8:
I agree that calculating a cost function based on student self-reported preferences is a fairly crude way to go about this, especially if the goal is to develop a system that can be used with many students at a time without having to first collect preferences using a survey. A more scientific way to go about this would be to give the questions as a pilot test, and rank the questions by relative difficulty, or use previously piloted questions which have empirically-derived estimations of difficulty. The idea is that more advanced students will be more likely to answer more difficult questions correctly. Providing the pilot dataset is representative, the difficulty-based costs could be generalized across all students. Wright and Stone’s book Best Test Design addresses these types of problems.
Response:
Indeed, there are more complex methods to create the student cost function. However, answering questions does not always represent the belief of the student regarding his skills. In some cases, students' preferences are disconnect from the student’s performance. Hence we ask the student about her preferences and did not used for example her grades to determine her preferences.
As specified in the future work section, we intend to explore more complex methods for creating cost functions that can be learned and become more accurate during the e-learning game after observing the students’ decisions in real time.
Reviewer 3 Report
The article formalizes the knowledge transfer process in the pedagogical setting as a multi-agent planning problem. The model is used to personalize learning content for math students. The formal description of the model is presented. The model is validated experimentally using a group of school students playing an e-learning game.
- The title of the article does not correctly represent the content of the paper and I suggest to revise it.
- The Didactic innovation is not clear. Serious games and gamification are already used in the education process widely. The novelty and contribution of this paper should be explicitly stated in the introduction section.
- The related work section should also address other formal or semiformal approaches to the modelling of teaching/learning process e.g. information theory based such as Towards empirical modelling of knowledge transfer in teaching/learning process.
- There is not much information about the methodology related to the research. There is a gap between the formal description of the method and its practical application for education.
- Section 8 describes the developed e-learning game. The authors should give more details of the methodology of game development. How is your game different from other games known in the scientific literature such as An interactive serious mobile game for supporting the learning of programming in JavaScript in the context of eco-friendly city management? While the game is the focus, the article does not explain the game scenario and gamification techniques employed to support player engagement and motivation to play.
- Describe game mechanics of the e-learning game, use the Machinations diagrams https://machinations.io/
- Explain how you maintain the interest in the game. What specific elements of gamification (badges, etc.) did you use to retain student motivation and interest in playing?
- Present more demographical information about game users (age, gender distribution).
- Table 4: add units of measurement (3rd column).
- Present a critical discussion section and discuss the limitations of the serious game for education as well as any threats to the validity of the results. Make a connection between your results and what were the results of previous similar studies, and the new and unique thing that you created. Also explicitly say what we learn from your work.
- Conclusions should be improved as it lacks insights and recommendations for further research in this domain. The claims are not supported. Use statistical analysis results to support the claims on the usefulness of the game.
Author Response
Dear reviewer, we thank you for reviewing our paper. We appreciate the detailed review and the many suggestions for improvement. We have addressed the issues raised by you, and the modifications to our manuscript. For your convenience the changes in the manuscript from previous submission were highlighted in blue.
Comment 1:
The title of the article does not correctly represent the content of the paper and I suggest to revise it.
Response:
Thanks for your suggestion. We feel that the title Supervisor Worker Problems and Applications in Education, captures well the work in the paper. Can you please expand on what is not correctly represented?
________________________________________________________________
Comment 2:
The Didactic innovation is not clear. Serious games and gamification are already used in the education process widely. The novelty and contribution of this paper should be explicitly stated in the introduction section.
Response:
We agree that Serious Games are well studied in education. Our novelty is not in creating a new serious game but in the mapping of a SWOPP model to a game design that can be shown to advance students’ learning without hindering their motivation. Manually constructing such a game for each student is not trivial task, and SWOPP provides an automatic way to do this.
________________________________________________________________
Comment 3:
The related work section should also address other formal or semiformal approaches to the modelling of teaching/learning process e.g. information theory based such as Towards empirical modelling of knowledge transfer in teaching/learning process.
Response:
Thank you for this suggestion, we add a discussion for this paper.
________________________________________________________________
Comment 4:
There is not much information about the methodology related to the research. There is a gap between the formal description of the method and its practical application for education.
Response:
Thanks for pointing this out. We added a sub section (8.1) describing the translation from SWOPP to the practical application for education.
________________________________________________________________
Comment 5:
Section 8 describes the developed e-learning game. The authors should give more details of the methodology of game development. How is your game different from other games known in the scientific literature such as An interactive serious mobile game for supporting the learning of programming in JavaScript in the context of eco-friendly city management? While the game is the focus, the article does not explain the game scenario and gamification techniques employed to support player engagement and motivation to play.
Response:
For us, the game is not the focus of this paper, but rather the supervisor-worker problem, the SWOPP formalization, and the ICFM algorithm. We consider the game to be a demonstration of the SWOPP model applied in real world problems.
We added more details about the game in section 8. Thanks for the citation.
________________________________________________________________
Comment 6:
Describe game mechanics of the e-learning game, use the Machinations diagrams https://machinations.io/
Response:
We added a schematic representation of the game flow, similar to the schematic diagram in the scientific literature you recommended An interactive serious mobile game for supporting the learning of programming in JavaScript in the context of eco-friendly city management
________________________________________________________________
Comment 7:
Explain how you maintain the interest in the game. What specific elements of gamification (badges, etc.) did you use to retain student motivation and interest in playing?
Response:
Thank you for your suggestion. We added details about the elements of gamification in section 8.
________________________________________________________________
Comment 8:
Present more demographic information about game users (age, gender distribution).
Response:
All students are from K5, hence no age distribution. We did not collect demographic information due to privacy constraints.
________________________________________________________________
Comment 9:
Table 4: add units of measurement (3rd column).
Response:
We removed this column from the paper.
________________________________________________________________
Comment 10:
Present a critical discussion section and discuss the limitations of the serious game for education as well as any threats to the validity of the results. Make a connection between your results and what were the results of previous similar studies, and the new and unique thing that you created. Also explicitly say what we learn from your work.
Response 10:
We revised the user study and results section to be clearer.
_______________________________________________________________
Comment 11:
Conclusions should be improved as it lacks insights and recommendations for further research in this domain. The claims are not supported. Use statistical analysis results to support the claims on the usefulness of the game.
Response:
We revised the user study and results section as well as the conclusion to be clearer.
Round 2
Reviewer 2 Report
The authors addressed a majority of my comments, with the exception of Comment 7. The lack of validated surveys for interest limits the ability of the authors to make claims around interest. I understand that is not necessarily the main contribution of the study, but this limitation could be noted in the section discussing limitations and future work. I believe being transparent about the limitations of measures used strengthens a work in that it lays a clear path for future work.
Author Response
Dear reviewer, we thank you again for reviewing our paper.
Comment:
The lack of validated surveys for interest limits the ability of the authors to make claims around interest. I understand that is not necessarily the main contribution of the study, but this limitation could be noted in the section discussing limitations and future work. I believe being transparent about the limitations of measures used strengthens a work in that it lays a clear path for future work.
Response:
We agree and have noted this limitation in the conclusion and future work section.
Reviewer 3 Report
The authors did great work in improving their paper. The application of the proposed method is explained more clearly now as the details on the mapping the game to the proposed model, the game flow, elements of gamification, and the experimental protocol have been added.
However, I still think that the title of the paper could still be improved as only one case study (application), rather than many applications, is analyzed and discussed in this paper. The problem of city planning mentioned in the abstract of the paper is not analyzed in this paper, so it could be removed from the abstract altogether.
Author Response
Dear reviewer, we thank you again for reviewing our paper.
Comment 1:
I still think that the title of the paper could still be improved as only one case study (application), rather than many applications, is analyzed and discussed in this paper
Response:
We changed the title to "Supervisor-Worker Problems with an Application in Education"
________________________________________________________________
Comment 2:
The problem of city planning mentioned in the abstract of the paper is not analyzed in this paper, so it could be removed from the abstract altogether
Response:
We removed this part from the abstract section.